# Toxic Effects of Amanitins: Repurposing Toxicities toward New Therapeutics

**DOI:** 10.3390/toxins13060417

**Published:** 2021-06-11

**Authors:** Brendan Le Daré, Pierre-Jean Ferron, Thomas Gicquel

**Affiliations:** 1Univ Rennes, INSERM, INRAE, CHU Rennes, Institut NuMeCan (Nutrition, Métabolismes et Cancer), Previtox Network, F-35000 Rennes, France; pierre-jean.ferron@inserm.fr (P.-J.F.); thomas.gicquel@chu-rennes.fr (T.G.); 2Centre Hospitalier Universitaire de Rennes, service Pharmacie, F-35000 Rennes, France; 3Centre Hospitalier Universitaire de Rennes, Laboratoire de toxicologie biologique et médico-légale, F-35000 Rennes, France

**Keywords:** amanitin, hepatotoxicity, *Amanita phalloïdes*, antidote, amanitin-conjugated antibody

## Abstract

The consumption of mushrooms has become increasingly popular, partly due to their nutritional and medicinal properties. This has increased the risk of confusion during picking, and thus of intoxication. In France, about 1300 cases of intoxication are observed each year, with deaths being mostly attributed to *Amanita phalloides* poisoning. Among amatoxins, α- and β-amanitins are the most widely studied toxins. Hepatotoxicity is the hallmark of these compounds, leading to hepatocellular failure within three days of ingestion. The toxic mechanisms of action mainly include RNA polymerase II inhibition and oxidative stress generation, leading to hepatic cell apoptosis or necrosis depending on the doses ingested. Currently, there is no international consensus concerning *Amanita phalloides* poisoning management. However, antidotes with antioxidant properties remain the most effective therapeutics to date suggesting the predominant role of oxidative stress in the pathophysiology. The partially elucidated mechanisms of action may reveal a suitable target for the development of an antidote. The aim of this review is to present an overview of the knowledge on amanitins, including the latest advances that could allow the proposal of new innovative and effective therapeutics.

## 1. Introduction

In recent years, the consumption of mushrooms has become increasingly common, probably due to their nutritional and medicinal properties. The risk of confusion when picking mushrooms, and therefore of intoxication, has therefore increased [1]. Worldwide, it is estimated that 10 to 50 of the 14,000 known species of mushrooms are potentially lethal [2,3]. Depending on their geographical distribution, the mushroom poisoning issue therefore varies. It represents a serious public health problem in many countries, including North America, Bulgaria, the Czech Republic, China, Iran, Mexico, Italy, Hungary, Japan, Nepal, Poland, Romania, Russia, South Korea, Thailand, Turkey, and Ukraine [2]. In France, data from poison centers report 10,662 mushroom poisonings between 2010 and 2017, with an increasing number since 2016. Of the 1300 cases per year on average, about 500 occur in October [4]. Because of the family gathering contexts, the age dispersion remains very wide (9 months–96 years) with a mean age of 45.2 years, and a sex ratio of 1. Of the 11 to 44 severe cases observed in our territory each year, an amanitin poisoning is found in 62.1% of cases, a pantherin syndrome in 12.6% of cases, and a sudorian muscarinic syndrome in 10.1% of cases. Up to five deaths per year are finally observed each year, mostly attributed to amanitin poisoning, and caused by the action of powerful toxins known as amanitins [4]. The aim of this review is to present an overview of the knowledge on amanitins, including the latest advances that could allow the proposal of new innovative and effective therapeutics.

In this review, MEDLINE and PubMed databases were searched for relevant papers published in English between 1960 and 2021 using the following search terms: amanitin, amanita, antidotes, toxicity, toxicokinetics, liver, hepatocyte, hepatotoxicity, RNA polymerase, silibinin, penicillin. Studies providing information about mechanistic explanation for amanitin toxicity, amanitin poisoning management, and amanitin-conjugated antibodies were included for review.

## 2. Amanitins

Amanitins are members of the amatoxin family, toxic bicyclic octapeptides with molecular weights of about 900 g/mol, contained in certain fungi. Currently, three major families of fungi are known to contain these toxins: the amanita (*Amanita* sp.), the galerina (*Galerina* sp.), and the lepiota (*Lepiota* sp.). Among all these species, the death cap (*Amanita phalloides*) is responsible for most of the fatal poisonings, the severe hepatic damage it causes being rapidly life-threatening [5] (Figure 1A).

This species is largely predominant in Europe, especially in Central and Western Europe [6]. However, cases of intoxication have also been described in North, Central, and South America; Australia; Asia; and Africa [7]. The amatoxin family is composed of at least nine compounds classified into neutral substances (α-amanitin, γ-amanitin, amaninamide, amanulline, and proamanulline), and acidic substances (β-amanitin, ε-amanitin, amanine, and amanullinic acid), without these differences in properties having been associated with a difference in toxicity [7]. These toxins are very soluble in water, and present a great resistance to heat and cold, making them extremely resistant to various consumption processes (cooking, freezing, drying) [8]. Moreover, these toxins are resistant to enzymes and acid degradation, and do not undergo any metabolism [9], making them resistant to intestinal inactivation and detoxification processes [7,8]. In parallel to amatoxins, two other families of toxins have been described as involved in the toxicity of these fungi: phallotoxins and virotoxins. However, given their non-absorption by the oral route, and the severity of the syndrome related to liver toxicity, only the amanitins are currently detailed.

### 2.1. Toxicokinetics of Amanitins

Among the amatoxins, α- and β-amanitin are the most widely studied (Figure 1B,C). Toxicokinetic data, obtained in animals and during human intoxications, report a very good absorption of amatoxins in the gastrointestinal tract and a detection in urine as early as 90 to 120 min post-ingestion [10,11,12]. The rapid distribution at the hepatic and renal level is explained in particular by a lack of binding to plasma proteins. After intravenous administration in dogs, the half-life of amatoxins has been estimated to be between 26.7 and 49.6 min, and their detection in blood is no longer possible after 4 to 6 h [10].

The liver is a privileged target of these toxins, receiving a massive amount of amanitins after gastrointestinal absorption, and specifically expressing the main transporter of these toxins at the sinusoidal level, OATP1B3 [13]. Moreover, the bile acid transporter NTCP has been identified as the second major transporter of amanitins in the liver [14]. Recently, our team showed an absence of α-amanitin metabolism in human both in vivo and in vitro using high resolution mass spectrometry coupled to molecular networking [9,10,11,12,13,14,15]. Amatoxins are excreted in large quantities in the urine (80 to 90% of the dose is found unchanged) during the first 72 h of intoxication [11]. Faulstich et al. (1985) also showed that a small amount of toxins could be eliminated by the bile (about 7%) and reabsorbed at the intestinal level, completing an enterohepatic cycle. Moreover, these same authors reported a fecal elimination of α-amanitin in the first 24 h [10]. Lastly, amatoxins undergo urinary elimination, the concentrations in the kidney cells being found 6 to 90 times higher than in the liver. This phenomenon explains in particular the nephrotoxicity reported within the framework of these intoxications, although dehydration due to gastrointestinal losses may also contribute to kidney injury [11,16].

### 2.2. Toxicity of Amanitins: From Clinical to Molecular Mechanisms

The primary toxic mechanism of amanitins is attributed to non-covalent nuclear inhibition of RNA polymerase type II (RNAP II), thereby decreasing mRNA levels and protein synthesis [17]. In parallel, this enzymatic inhibition has been shown to be responsible for the ubiquitination of RNAP II and its degradation by the proteasome, correlated with an increase in intracellular ATP concentrations [18,19] (Figure 2). With the liver being a major player in protein synthesis, it is therefore more impacted. The stress signals induced by amanitins, including RNA polymerase inhibition, lead to an induction of the p53 protein, allowing for the formation of complexes with anti-apoptotic proteins (Bcl-XL and Bcl-2) and the triggering of apoptosis by mitochondrial release of cytochrome c in the cytosol [20,21,22]. Wang et al. (2018) showed that α-amanitin induced significant changes in the mitochondrial proteome, associated with destruction of membrane potential [23]. Thus, amanitin-induced apoptosis has been proposed to be critical in the pathophysiology of these intoxications [24] (Figure 2). The generation of oxidative stress has also been suggested to be important in the development of these severe hepatotoxicities. Indeed, it has been shown that α-amanitin accumulation leads to an increase in superoxide dismutase (SOD) and glutathione peroxidase activity, malondialdehyde products, and lipid peroxidation, correlated with an inhibition of catalase activity [25,26]. Recently, new mechanistic data have shown that amanitins induce the production of GSH and tGSH, reinforcing the hypothesis of the involvement of oxidative stress in this pathophysiology [18]. Zheleva (2013) showed that α-amanitin was able to form phenoxyl-free radicals that may be involved in the increased production of reactive oxygen species (ROS) [27] (Figure 2). Lastly, an induction of the NF-κB pathway in the case of amanitin intoxication has been shown, conferring a certain protective effect, without a link being established with the levels of production of SOD, GSH, or catalase [28,29]. There are still many gray areas regarding the toxic effects of amanitins at the cellular level, which explains the absence of highly effective specific therapies. To this day, the treatment remains mainly symptomatic.

As mentioned previously, the symptomatology of α-amanitin intoxication is overwhelmingly attributed to its accumulation in the liver and kidney. However, symptoms only occur when significant damage is done. Therefore, clinical signs are only noticeable several hours to several days after ingestion. In the literature, three phases are classically distinguished: (i) the gastrointestinal phase, (ii) the latency period, and (iii) the hepato-renal phase [30]. The first phase appears abruptly, 6 to 24 h after ingestion, and is characterized by nausea, vomiting, diarrhea, abdominal pain, and hematuria. It lasts 12 to 36 h, and may be accompanied by fever; tachycardia; and metabolic disorders such as hypoglycemia, dehydration, and electrolyte disturbances [6]. After this initial phase, a deceptively reassuring period of asymptomatic latency occurs, during which hepatorenal damage progressively sets in, approximately 72 h after ingestion [7,31]. Lastly, when liver damage becomes significant, the third phase is characterized by an increase in transaminases (ALT/AST) and lactate dehydrogenase (LDH), as well as an alteration in coagulation by a decrease in the prothrombin level. The appearance of hemorrhagic manifestations by inhibition of hepatic synthesis of coagulation factors, encephalopathy, or cardiomyopathy will constitute elements of bad prognosis [7,31]. In cases of fatal outcome, the median time to death was 6.1 days (2.7–13.9 days) [32]. Acute kidney injuries frequently observed during these intoxications can be attributed to the nephrotoxic action of amanitins, to the gastrointestinal losses, and to the previous state of kidneys.

The hepatic histological data described in the context of amanitin poisoning vary according to the severity of the intoxication. In a series of eight cases of moderate non-fatal poisoning, Wepler et al. (1972) reported the appearance of centrilobular necrosis without steatosis or development of an initial inflammatory response. After several weeks, phagocytosis of the necrotic cells by Kupffer cells activated a mild inflammatory response associated with infiltration of lymphocytes [33]. In the case of fatal intoxications, the presence of steatosis has been described as preceding the appearance of centrilobular necrosis. In addition, three phases have been described, showing (i) the penetration of erythrocytes into the hepatocytes, (ii) the appearance of lysosomal fusions, and (iii) the development of centrilobular necrosis associated with hepatic hemorrhagic infiltrates [34]. In mice, electron microscopy studies have shown very early hepatocyte changes after administration of α-amanitin. As early as 30 min after administration of this toxin, there was a decrease in the number of perichromatin fibrils, associated with nuclear fragmentation and an increase in perichromatin and interchromatin granules from 1 to 5 h after the start of treatment [35].

## 3. Amanitin Poisoning Management

Currently, there are no authoritative international recommendations for the management of *Amanita phalloides* poisoning. However, general principles have been described in the literature. During the gastrointestinal phase, electrolyte imbalances and dehydration may occur, which may require stabilization of vital functions, including the administration of fluids. Moreover, inhibition of protein synthesis and hepatocyte damage may impair the production of coagulation factors, and thus intravenous administration of fresh frozen plasma may be necessary [36]. As the excretion of these toxins is predominantly urinary, forced diuresis has been suggested as a means of increasing their clearance, and is recommended for 4 to 5 days, with a target diuresis of 100 to 200 mL/h. Extracorporeal purification techniques such as hemodialysis, hemoperfusion, or plasmapheresis have been described, although their effectiveness remains limited in view of the rapid blood clearance of amanitins [11]. However, in cases of acute liver failure, treatment with high-volume plasma exchange has been shown to increase liver transplant-free survival [37]. Beyond a potential detoxifying treatment, high-volume plasma exchange would also allow the removal of danger-associated molecular patterns (molecules within cells that are a component of the innate immune response released from damaged or dying cells) involved in liver failure [37]. Interestingly, Sun et al. showed that interruption of the enterohepatic cycling of amatoxins by biliary drainage in dogs caused a more than 70% reduction in intestinal amatoxin absorption. In addition, the dogs with biliary drainage showed less severe toxicity signs and biochemical and pathological changes and much lower internal exposure than dogs without biliary drainage. These results suggest a promising additional therapy in case of *Amanita phalloides* intoxication [38]. On the drug side, many substances have been tested as potential antidotes, mostly hormones (insulin, growth hormone, glucagon), steroids, vitamin C, vitamin E, cimetidine, α-lipoic acid (also called thioctic acid), antibiotics (benzylpenicillin, ceftazidime), N-acetylcysteine, and silibinin and silymarin [31,39]. Although the pathophysiological basis is only partially understood, (i) inhibition of the OATP1B3 entry transporter (by β-lactam core antibiotics), (ii) induction of antioxidant effects (by N-acetylcysteine, vitamin C, vitamin E, cimetidine, α-lipoic acid), or (iii) a combined antioxidant and OATP1B3 inhibitory effect (by silibinin or silymarin) are therefore the main therapeutic approaches available [32,39,40].

On the basis of a retrospective study of 2108 patients hospitalized for *Amanita phalloides* poisoning in North America and Europe between 1971 and 2001 (overall mean mortality of 11.6%), Poucheret et al. (2010) evaluated the effectiveness of these different treatments [39,41]. Although no randomized controlled trials have been performed to date regarding the management of *Amanita phalloides* poisoning, the studies reported here provide patient survival data associated with various drug classes after multidimensional statistical analysis. Thus, although no high level of evidence recommendation can be established yet, it is interesting to observe the survival trends of patients to propose an empirical therapeutic strategy. These authors concluded that silibinin (alone or in combination) and N-acetyl-cysteine should be used as first-line therapy, due to the observed mortality rates of 5.6% and 6.8%, respectively. Accordingly, a mixture of silibinin A and B (flavonolignan obtained from the extraction of the seeds from milk thistle fruit) has been approved for the treatment of *Amanita phalloides* intoxication in the European Union under the name Legalon^®^ (Rottapharm Madaus, Cologne, Germany) [42]. From experimental results, silibinin showed interesting mechanisms of action, in addition to the competitive inhibition of OATP1B3 and the decrease of oxygenated free radical production and lipid peroxidation [13]. It appears that silibinin may inhibit TNF-α release in injured hepatocytes, thus decreasing TNF-α-induced apoptosis in amatoxin poisoning. In addition, by increasing the number of ribosomes in the cell and therefore protein synthesis, silibinin can enhance the regenerative capacity of the liver [42]. Lastly, silibinin may also exert its hepatoprotective effects by its anti-inflammatory and anti-fibrotic effects and the inhibition of the binding of amatoxin to hepatocyte membranes [42]. N-Acetyl-cysteine, which is indicated in patients with fulminant liver failure, must be recommended in this context. Moreover, although benzylpenicillin alone or in combination is the most widely reported treatment in the literature, this treatment option does not show any trend towards improvement or reduction in mortality rates (10.7%), highlighting a lack of demonstrated efficacy and leading to question its use. Vitamin E, other antibiotics (gentamycin, neomycin, streptomycin, vancomycin, clindamycin), vitamin C, cimetidine, and thioctic acid were associated with mortality rates of 40%, 20.3%, 19%, 14.3%, and 12%, respectively, showing no influence on patient survival. Lastly, the use of steroids, growth hormones, and glucagon was associated with a decrease in patient survival, leading to the exclusion of these treatments from the management of these intoxications [39,41]. Apart from considerations of mechanism of action, it seems prudent to rule out the use of vancomycin or aminoglycosides in these patients at high risk of renal failure because of the nephrotoxicity of these molecules. Table 1 summarizes the different clinical efficacy data of the main antidotes used in the management of *Amanita phalloides* poisoning in humans. Overall, although randomized control studies are lacking, penicillins or similar, along with silibinin agglutinate the most positive reports, and other therapies have not pointed at potential beneficial results in case series. In parallel, new therapeutic combinations seem to emerge and deserve further preclinical investigations; for example, polymyxin B inhibiting the OATP1B3 and RNAP II transporter and methylprednisolone inhibiting the NTCP transporter—this combination has been proposed as a potential new antidote, due to positive results in mice [43]. However, further studies would be necessary to propose this combination in humans because of the negative impact of corticosteroids observed in real life.

Similar to other intoxications such as digitalis [44], immunotherapy against amanitins has been proposed in the context of *Amanita phalloides* poisoning management. However, although a partial protective effect of these antibodies against amanitins was demonstrated in vitro in Chang cells [45], strong nephrotoxicity associated with amanitin-specific antibodies found in mice did not lead to further development [46,47]. Recently, M101 (an extracellular hemoglobin extracted from the marine worm *Arenicola marina*) has been used as an effective antidote to reduce amanitin-induced cell death and mitochondrial reactive oxygen species production in hepatocyte-like HepaRG cells, leading to the filing of a patent [48]. Although these results have only been supported in vitro, they represent a therapeutic hope in the management of *Amanita phalloides* poisonings.

## 4. Perspectives

In view of the mechanisms of toxic action of amanitins described above, it seems that the pathophysiology of liver damage is not fully understood. This observation partly explains the absence of a highly effective antidote on the market and thus constitutes an ongoing challenge in toxicology. Recently, Popp et al. (2020) suggested that the use of 3D models (spheroids and organoids) would be particularly relevant in the study of the efficacy and safety of new antidotes, opening up interesting perspectives [49]. From a general standpoint, simultaneous presence of different bioactive constituents in hepatotoxic mushrooms is also potentially able to produce synergistic toxic effects that cannot be predicted from the toxicity of individual mycotoxins. Since the pathophysiology of *Amanita phalloides* intoxications has been mainly focused on amanitins, it seems imperative to investigate the presence of other toxins. Accordingly, the use of “-omics” analytical methods could of particular interest and might lead to the discovery of effective specific antidotes. In this context, fundamental research remains essential in the advancement of knowledge, allowing us to hope for the discovery of tomorrow’s effective antidotes.

Interestingly, the toxic effects of α-amanitin are beginning to be exploited for therapeutic purposes in oncology [50]. In particular, the achievement of the total synthesis of α-amanitin in recent years has allowed for the development of new amanitin-conjugated antibodies targeting certain tumor characteristics [51,52]. Since α-amanitin is no longer a substrate for OATP1B3 when coupled to antibodies, the particularities of the conjugated antibody allow targeting a precise cell population while limiting the non-selective toxicity [52]. Accordingly, various studies using α-amanitin-conjugated anti-epithelial cell adhesion molecule (EpCAM) antibody in murine models suggest their interest in colorectal cancer, pancreatic carcinomas, and various EpCAM-expressing malignancies [53,54,55]. Moreover, HDP-101 (an anti-B-cell maturation antigen (BCMA) antibody conjugated with an amanitin derivative) displayed high efficacy against both proliferating and resting myeloma cells in vitro, sparing BCMA-negative cells. These findings might provide a novel therapeutic approach to overcome drug resistance in this disease [56]. Recently, Gallo et al. (2021) showed that α-amanitin-based small-molecule drug conjugates grafted onto an immunoglobulin Fc domain increased the pharmacokinetic properties and therapeutic efficacy over the small-molecule drug conjugates in a prostate cancer xenograft mouse model [57]. Overall, recent progress in the knowledge of amanitins could make it possible to take advantage of their toxic effects, offering promising therapeutic perspectives.

## Figures and Tables

**Figure 1 toxins-13-00417-f001:**
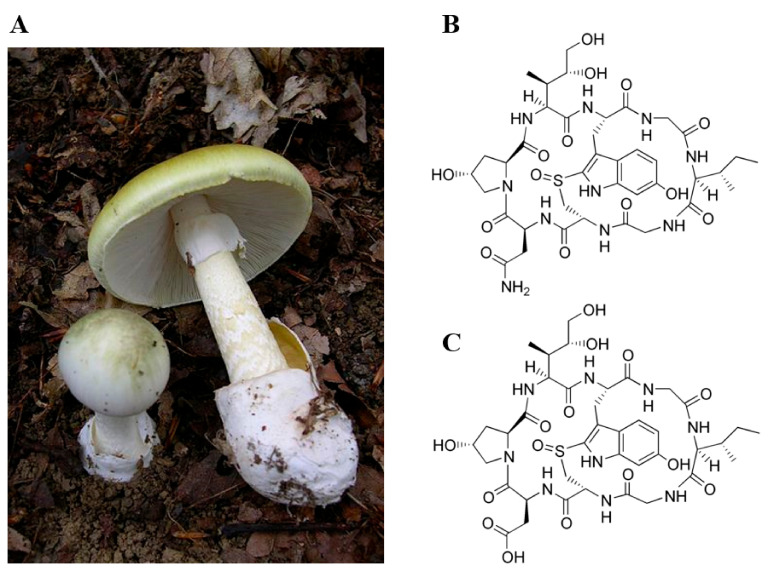
(**A**) *Amanita phalloides.* Visualization of the characteristic elements gathering an olive-green color of the cap, a broad body, a volva in bag, a soft ring in skirt, and numerous and free white blades leaving a white spore. (**B**) Molecular structure of α-amanitin. (**C**) Molecular structure of β-amanitin.

**Figure 2 toxins-13-00417-f002:**
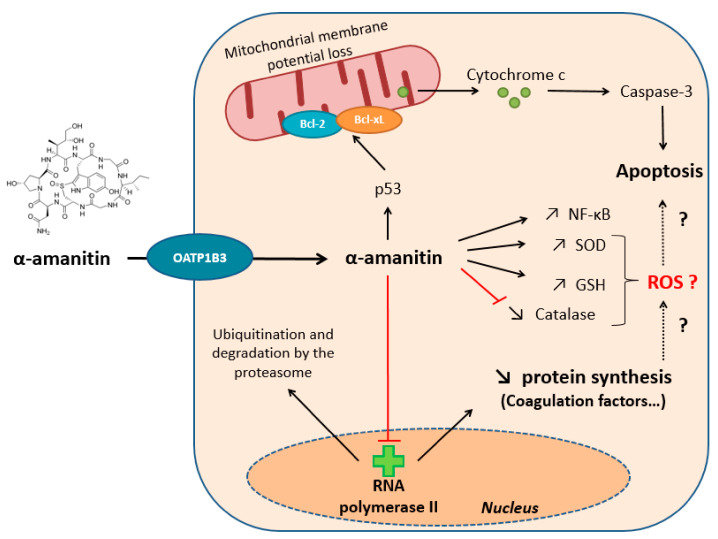
Main toxic mechanisms of amanitins within hepatocytes.

**Table 1 toxins-13-00417-t001:** Clinical efficacy data of the main antidotes used in the management of *Amanita phalloides* poisoning in humans. Patient survival rates are associated with various drug classes after multidimensional statistical analysis.

Empiric Therapeutic Strategy	Molecules	Putative Mechanism of Action	Associated Mortality Rate (11.6% Average Mortality)	References
**First-line**	Silibinin (alone or in combination) (n = 624)	Inhibitor of the OATP1B3 transporter Antioxidant effectsAnti-inflammatory effects Tissue repair	5.6%	[39,41]
	N-acetyl-cysteine (n = 192)	Antioxidant effects	6.8%	
	Ceftazidime (combined with silibinin) Positive impact on a small number of patients (n = 12); interest to be demonstrated on larger samples	OATP1B3 transporter inhibitor	0%	
**Second-line** If first-line treatments are not available	Benzylpenicillin alone or in combination (n = 1411)		10.7%	
**Interest still to be demonstrated** No positive impact on care or deleterious impact on patient survival observed in a small number of patients	Vitamin E (n = 25)	Antioxidant effects	40%	
	Vitamin C (n = 60), cimetidine (n = 21), thioctic acid (n = 450)		12–20.3%	
	Gentamycin, neomycin, streptomycin, vancomycin, clindamycin (n = 63 in the entire group)	Unknown		
	Insulin + growth hormone (n = 69), insulin + glucagon (n = 128)	Stimulation of the hepatic metabolism	16%	
	Steroids (n = 459)	Anti-inflammatory		

## Data Availability

No new data were created or analyzed in this study. Data sharing is not applicable to this article.

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
