# Peer review of "Toxic Effects of Amanitins: Repurposing Toxicities toward New Therapeutics"

_toxins, 2021, doi:10.3390/toxins13060417_

Round 1

Reviewer 1 Report

General Comments:

The manuscript was generally well written and provided substantial information about amanitins, particularly with respect to mechanisms of toxic action, clinical effects and therapeutic approaches to treating intoxications.  The title was perhaps somewhat confusing in that the discussion on the use of amanitin as a chemotherapeutic was unexpected and was not suggested in the abstract.  The description of different interventions to treat intoxication was generally comprehensive, but unfortunately, more superficial than might have been desired and most were not reviewed critically.  There were some statements that suggest the authors did not have input from a physician offering a critique of the treatments provided (see specific comments below).

References were in two different formats – use required journal format consistently.

Specific comments:

Title:  consider a more descriptive title

Introduction: no mention of North America in list of areas/countries with amanitin intoxications.

Page 1, line 32:  perhaps replace sudorian with muscarinic – not sure how many readers will know what the term sudorian means.

Page 2, 2nd paragraph under Toxicity of amanitins:  to this reviewer it makes more sense to describe the toxic mechanisms before describing the pathological lesions.

Page 3, line 109 – the erythrocytes do not penetrate into the hepatocytes – please clarify.

Page 2, line 120:  Figure number is mislabeled – here and Figure 3 (there is no figure 3).

Page 2-3: beginning line 120 – the actinomycin and acridine comments seem disconnected to the discussion of amanitins in terms of molecular targets – their mechanisms are different and not sure of the relevance in the context of the review.

Page 4, lines 142-144: the discussion of NF-kB pathway does not appear to align with the Figure showing various cellular events – the figure shows NF-kB to increase implying a positive effect on ROS production but the description does not support this.

Page 4, lines 144-146:  the comment about “absence of specific therapies” is misleading – the authors describe many specific therapies – unfortunately most are not effective.

Page 5: the discussion of case management and treatment of amanitin intoxication is weak from the standpoint of a critical review of various treatments that have been tried.  For example, line 156 suggests vitamin K as being necessary. However, if one understand the effect of amanitin in decreasing coagulation factor production and how vitamin K works, there does not appear to be any usefulness to giving vitamin K at all.  The discussion of decontamination demonstrates a lack of knowledge with regard to current thoughts on decontamination – that gastric lavage is rarely recommended and aspiration of ingested contents or surgical removal is not correct.

Page 5, line 171 – should AOTP1B3 be OATP1B3?

Page 5, beginning line 176 – the discussion of silibinin is inadequately discussed given the intent of the paper – there is an approved drug in the EU (Legalon) that is marketed as an effective treatment and given the percentages listed regarding lowered mortality rates deserves more discussion.

Page 5, line 181 – the statement that benzylpenicillin (misspelled here) is the most widely used treatment either alone or in combination is misleading – emphasis should be placed on its lack of demonstrated efficacy.

Page 5, line 192 – should ARNP II be RNAP II?  Please address the apparent discrepancy between the comment about the potential for a steroid (methylprednisolone) to be effective but saying that steroids are deleterious in Table 1.

Table 1:  Silybine – is this intended to be silybin?  0% mortality for ceftazidime – is this correct? Formatting seems off after 2nd line.  Again, the authors don’t critically evaluate why the therapeutics listed as deleterious and actually deleterious. 

As a current review, the authors might wish to explore a current theory that gall bladder cannulation can improve case outcomes by interrupting enterohepatic circulation. 

Page 6, lines 197-200 – Chen et al. (reference 45) did not do any in vivo work so this is misleading – in fact Chen suggested some potential benefit.  It would be useful to discuss more generally why immunotherapy is not potentially useful (i.e., why hasn’t it been pursued since the Chen paper in 1993?

Page 6, Perspectives – the first paragraph, in this reviewer’s opinion, does not add anything of note to the review.  When referring to “mycotoxins” are the authors referring to other bioactive constituents in hepatotoxic mushrooms (e.g., phallotoxin) or are they referring to mycotoxins more generally (e.g., aflatoxins)?  Interactions can be inhibitory, additive, or synergistic – using only synergistic might or might not be correct.

The discussion of using amanitin as a chemotherapeutic seems out of place under “Perspectives” – also refer back to comment about the title of the review. 

Reference 15 is missing information.

Author Response

4th June 2021

Editorial

Toxins

Dear Editor,

Please find enclosed our revised manuscript entitled: “Toxic effects of amanitins: repurposing toxicities toward new therapeutics.” as a submission for publication in Toxins.

We would like to thank you for your helpful and constructive feedback concerning our manuscript. We discussed all the comments and - where appropriate - amended the manuscript accordingly as indicated. For better clarification, please find enclosed a reply letter that addresses all of the comments. In the manuscript, we highlighted all changes in red.

This work has not been nor will be submitted for publication in any other journal until you have taken your final decision.

We hope that you will find it suitable for publication.

Thank you for your cooperation.

Sincerely yours.

Reviewer #1:

The manuscript was generally well written and provided substantial information about amanitins, particularly with respect to mechanisms of toxic action, clinical effects and therapeutic approaches to treating intoxications.  The title was perhaps somewhat confusing in that the discussion on the use of amanitin as a chemotherapeutic was unexpected and was not suggested in the abstract. The description of different interventions to treat intoxication was generally comprehensive, but unfortunately, more superficial than might have been desired and most were not reviewed critically.  There were some statements that suggest the authors did not have input from a physician offering a critique of the treatments provided (see specific comments below).

References were in two different formats – use required journal format consistently.

Specific comments:

Title:  consider a more descriptive title.

Thank you for this comment. We then modified the title as suggested by reviewer 4, page 1 of the manuscript:

“Toxic effects of amanitins: repurposing toxicities toward new therapeutics.”

Introduction: no mention of North America in list of areas/countries with amanitin intoxications.

Thank you for this relevant remark. We thus added North America in list of other areas/countries with amanitin intoxications.

“It represents a serious public health problem in many countries, including North America, Bulgaria, the Czech Republic, China, Iran, Mexico, Italy, Hungary, Japan, Nepal, Poland, Romania, Russia, South Korea, Thailand, Turkey and Ukraine [2].”

Page 1, line 32:  perhaps replace sudorian with muscarinic – not sure how many readers will know what the term sudorian means.

This term has been changed page 1 of the manuscript.

Page 2, 2nd paragraph under Toxicity of amanitins:  to this reviewer it makes more sense to describe the toxic mechanisms before describing the pathological lesions.

Thank you for this comment. We thus described the toxic mechanisms before describing the pathological lesions, page 3 to 5 of the manuscript.

Page 3, line 109 – the erythrocytes do not penetrate into the hepatocytes – please clarify.

Indeed, these descriptions also surprised us. However, since we are not pathologists, we have adopted the exact terms used by the authors of the publication, which are:

“Histologic changes in the liver showed three stages. The first stage consisted of hepatocytes sometimes penetrated by erythrocytes and containing vacuoles with weakly eosinophilic granular content... “ (Fineschi V, Di Paolo M, Centini F. Histological Criteria for Diagnosis of Amanita Phalloides Poisoning. J Forensic Sci. 1 mai 1996;41(3):13929J.)

We thus decided to keep the message as close as possible to the researchers' descriptions.

Page 2, line 120:  Figure number is mislabeled – here and Figure 3 (there is no figure 3).

We apologize for this typo. We have corrected it.

Page 2-3: beginning line 120 – the actinomycin and acridine comments seem disconnected to the discussion of amanitins in terms of molecular targets – their mechanisms are different and not sure of the relevance in the context of the review.

We thus deleted the associated section in the manuscript.

Page 4, lines 142-144: the discussion of NF-kB pathway does not appear to align with the Figure showing various cellular events – the figure shows NF-kB to increase implying a positive effect on ROS production but the description does not support this.

Thank you for this relevant comment. We have modified the figure to separate the NF-κB pathway from the oxidative stress pathways, page 5 of the manuscript:

Page 4, lines 144-146:  the comment about “absence of specific therapies” is misleading – the authors describe many specific therapies – unfortunately most are not effective.

We thus modified the text, page 5 of the manuscript:

“There are still many grey areas regarding the toxic effects of amanitins at the cellular level, which explains the absence of highly effective specific therapies.”

Page 5: the discussion of case management and treatment of amanitin intoxication is weak from the standpoint of a critical review of various treatments that have been tried.  For example, line 156 suggests vitamin K as being necessary. However, if one understand the effect of amanitin in decreasing coagulation factor production and how vitamin K works, there does not appear to be any usefulness to giving vitamin K at all.  The discussion of decontamination demonstrates a lack of knowledge with regard to current thoughts on decontamination – that gastric lavage is rarely recommended and aspiration of ingested contents or surgical removal is not correct.

We thus modified the text page 6 of the manuscript:

“Also, inhibition of protein synthesis and hepatocyte damage may impair the production of coagulation factors, so intravenous administration of vitamin K and fresh frozen plasma may be necessary [35].”

Also, since digestive decontamination is indeed much discussed in the literature, we have removed the sentence referring to it, page 6 of the manuscript:

“The prevention of amatoxin absorption is possible in case of very early medical management (ideally < 1 h post-ingestion) by gastric lavage, administration of activated charcoal or aspiration of the ingested contents by endoscopic or surgical means. In addition, the administration of activated charcoal may allow the interruption of the enterohepatic cycle, limiting the accumulation of toxins in the liver (7).”

Page 5, line 171 – should AOTP1B3 be OATP1B3?

We apologize for this typo. We have corrected it.

Page 5, beginning line 176 – the discussion of silibinin is inadequately discussed given the intent of the paper – there is an approved drug in the EU (Legalon) that is marketed as an effective treatment and given the percentages listed regarding lowered mortality rates deserves more discussion.

We thus added a paragraph on silibinin, page 7 of the manuscript:

“Accordingly, a mixture of silibinin A and B (flavonolignan obtained from the extraction of the seeds from milk thistle fruit) has been approved for the treatment of Amanita phalloides intoxication in the European Union under the name Legalon® (Rottapharm Madaus, Cologne, Germany) [41]. From experimental results, silibinin showed interesting mechanisms of action, in addition to the competitive inhibition of OATP1B3 and the decrease of oxygenated free radical production and lipid peroxidation [13]. It appears that silibinin may inhibit TNF-α release in injured hepatocytes, thus decreasing TNF-α -induced apoptosis in amatoxin poisoning. In addition, by increasing the number of ribosomes in the cell and therefore protein synthesis silibinin can enhance the regenerative capacity of the liver [41]. Lastly, silibinin may also exert its hepatoprotective effects by its anti-inflammatory and anti-fibrotic effects and the inhibition of the binding of amatoxin to hepatocyte membranes [41].“

We have also homogenized the terms silibinin and silybin in the text, keeping only silibinin.

Page 5, line 181 – the statement that benzylpenicillin (misspelled here) is the most widely used treatment either alone or in combination is misleading – emphasis should be placed on its lack of demonstrated efficacy.

We thus modified and moderated the sentence, page 7 of the manuscript:

“Also, although benzylpenicillin alone or in combination is the most widely used reported treatment in the literature, this treatment option does not show any trend towards improvement or reduction in mortality rates (10.7%), highlighting a lack of demonstrated efficacy and leading to question its use.”

Page 5, line 192 – should ARNP II be RNAP II?  Please address the apparent discrepancy between the comment about the potential for a steroid (methylprednisolone) to be effective but saying that steroids are deleterious in Table 1.

Thank you for this comment. Table I is dedicated to clinical efficacy data in humans only. Since methylprednisolone in association with polymyxin B have been only used in mice, we aimed to separate these two pieces of information. We added a sentence to clarify it, page 7 of the manuscript:

“However, further studies would be necessary to propose this combinaison in humans, because of the negative impact of corticosteroids observed in real life.”

Table 1:  Silybine – is this intended to be silybin?  0% mortality for ceftazidime – is this correct? Formatting seems off after 2nd line.  Again, the authors don’t critically evaluate why the therapeutics listed as deleterious and actually deleterious. 

Since silibinin and silybin (or silybine) refer to the same substance, we have homogenized the terms silibinin and silybin in the text, keeping only silibinin.

Concerning Ceftazidime, the 0% mortality rate is correct but reports data from a single study performed on 12 patients. Because of this major limitation, we wished to specify it in Table I in order to take it into account for the interpretation of the data.

To simplify Table I, we also grouped the sections “interest still to be demonstrated” and “to be excluded from care” to only keep “Interest still to be demonstrated (no positive impact on care or deleterious impact on patient survival observed in a small number of patients)”.

As a current review, the authors might wish to explore a current theory that gall bladder cannulation can improve case outcomes by interrupting enterohepatic circulation. 

We thus added a paragraph on enterohepatic circulation interrupting, page 6 of the manuscript:

“Interestingly, Sun et al. showed that interruption of the enterohepatic cycling of amatoxins by biliary drainage in dogs caused a more than 70% reduction in intestinal amatoxin absorption. In addition, the dogs with biliary drainage showed less severe toxicity signs and biochemical and pathological changes and much lower internal exposure than dogs without biliary drainage. These results suggest a promising additional therapy in case of Amanita phalloides intoxication [37].

37 - Sun J, Zhang Y-T, Niu Y-M, Li H-J, Yin Y, Zhang Y-Z, et al. Effect of Biliary Drainage on the Toxicity and Toxicokinetics of Amanita exitialis in Beagles. Toxins 2018;10:215. https://doi.org/10.3390/toxins10060215.

Page 6, lines 197-200 – Chen et al. (reference 45) did not do any in vivo work so this is misleading – in fact Chen suggested some potential benefit.  It would be useful to discuss more generally why immunotherapy is not potentially useful (i.e., why hasn’t it been pursued since the Chen paper in 1993?

This is absolutely true. In vivo results found in mice revealed nephrotoxicity, jeopardizing the safety of use. To clarify our sentence, we modified the text page 7 of the manuscript and added the references of interest:

“Similar to other intoxications such as digitalis [43], immunotherapy against amanitins has been proposed in the context of Amanita phalloides poisoning management. However, although a partial protective effect of these antibodies against amanitins was demonstrated in vitro in Chang cells [44], strong nephrotoxicity associated with amanitin-specific antibodies found in mice did not lead to further development [45,46].”

[45]      Faulstich H, Kirchner K, Derenzini M. Strongly enhanced toxicity of the mushroom toxin α-amanitin by an amatoxin-specific Fab or monoclonal antibody. Toxicon 1988;26:491–9. https://doi.org/10.1016/0041-0101(88)90188-2.

[46]      Kirchner K, Faulstich H. Purification of amatoxin-specific antibodies from rabbit sera by affinity chromatography, their characterization and use in toxicological studies. Toxicon 1986;24:273–83. https://doi.org/10.1016/0041-0101(86)90152-2.

Page 6, Perspectives – the first paragraph, in this reviewer’s opinion, does not add anything of note to the review. When referring to “mycotoxins” are the authors referring to other bioactive constituents in hepatotoxic mushrooms (e.g., phallotoxin) or are they referring to mycotoxins more generally (e.g., aflatoxins)?  Interactions can be inhibitory, additive, or synergistic – using only synergistic might or might not be correct.

We have therefore clarified our remarks, page 8 and 9 of the manuscript:

“From a general standpoint, simultaneous presence of different bioactive constituents in hepatotoxic mushrooms mycotoxins is also potentially able to produce synergistic toxic effects that cannot be predicted from the toxicity of individual mycotoxins.”

The discussion of using amanitin as a chemotherapeutic seems out of place under “Perspectives” – also refer back to comment about the title of the review. 

We agree that the title is not optimal for the content of the manuscript. We have therefore modified the title as stated above to best represent the scope of the article (“Toxic effects of amanitins: repurposing toxicities toward new therapeutics.”).

Reference 15 is missing information.

We have completed the reference:

[15]      Mydlík M, Derzsiová K. Liver and kidney damage in acute poisonings. BANTAO Journal 2006, 4(1), 30.

We would like to thank you for your helpful and constructive feedback concerning our manuscript and we hope that you will find it suitable for publication.

Reviewer 2 Report

Manuscript#: toxins-1232842

Toxic effects of amanitins: toward new effective therapeutics

In the present manuscript authors aim to present an overview of the knowledge on amanitins, including the latest advances that could allow the proposal of new innovative and effective therapeutics. Authors cover the main aspects of α-amanitin toxicity and respective antidotes and briefly mention new effective anticancer therapeutics using new amanitin-conjugated antibodies. Although this manuscript does not provide particularly new or enthusiastic perspectives, the authors have developed good work in this area and this review will allow them to consolidate their contribution to the development of new therapies in the future.

Author Response

Reviewer #2:

In the present manuscript authors aim to present an overview of the knowledge on amanitins, including the latest advances that could allow the proposal of new innovative and effective therapeutics. Authors cover the main aspects of α-amanitin toxicity and respective antidotes and briefly mention new effective anticancer therapeutics using new amanitin-conjugated antibodies. Although this manuscript does not provide particularly new or enthusiastic perspectives, the authors have developed good work in this area and this review will allow them to consolidate their contribution to the development of new therapies in the future.

Thank you for your remarks. We modified the text as suggested by other reviewers. We hope you will find this version improved and suitable for publication.

Reviewer 3 Report

Attached doc

Author Response

4th June 2021

Editorial

Toxins

Dear Editor,

Please find enclosed our revised manuscript entitled: “Toxic effects of amanitins: repurposing toxicities toward new therapeutics.” as a submission for publication in Toxins.

We would like to thank you for your helpful and constructive feedback concerning our manuscript. We discussed all the comments and - where appropriate - amended the manuscript accordingly as indicated. For better clarification, please find enclosed a reply letter that addresses all of the comments. In the manuscript, we highlighted all changes in red.

This work has not been nor will be submitted for publication in any other journal until you have taken your final decision.

We hope that you will find it suitable for publication.

Thank you for your cooperation.

Sincerely yours.

Reviewer #3:

The review is overall very interesting and synthetic. The main problem related with the topic is that, in fact, there is no straight evidence for nothing we do with Amanita intoxication besides and a combination of “a priori” effective treatments; randomized controlled trials are lacking in such an uncommon and geographical dependent event. My comments regarding the review:

-I would expand the sections regarding animal mechanistic studies to add for the search of new therapies.

We have therefore added a paragraph on the enterohepatic circulation of amanitins in order to open perspectives on the search for new therapies, page 6 of the manuscript:

“Interestingly, Sun et al. showed that interruption of the enterohepatic cycling of amatoxins by biliary drainage in dogs caused a more than 70% reduction in intestinal amatoxin absorption. In addition, the dogs with biliary drainage showed less severe toxicity signs and biochemical and pathological changes and much lower internal exposure than dogs without biliary drainage. These results suggest a promising additional therapy in case of Amanita phalloides intoxication [37].”

-Plasmapheresis, plasma exchange or depurative techniques: I would dedicate a few more lines to these topics since being relatively used in this setting. They both would act as potential detoxifying treatments as well as depurative agents for DAMPS and other molecules implied in liver failure. It must be clarified, and cited by authors, that plasma exchange (Larssen, Journal of Hepatology) is the only intervention that has shown to increase transplant-free survival in an RCT, specially for patients with another known intoxication: paracetamol.

Thank you for this relevant remark. We thus added a sentence in this paragraph page 6 of the manuscript:

“However, in case of acute liver failure treatment with high volume plasma exchange, acting as potential detoxifying treatments as well as depurative agents for danger-associated molecular patterns and other molecules implied in liver failure, has been shown to increase liver transplant-free survival [36].”

-Last paragraph regarding new amantinin-based oncologic therapies. I would absolutely shorten this paragraph since exceeding the scope of the review, from my point of view is very speculative.

We thus shorten this paragraph, page 9 of the manuscript.

-AKI: not only related to direct toxin effect but to dehydration because of GI loses.

This is absolutely true. We thus clarified the paragraph, page 3 of the manuscript:

“Lastly, amatoxins undergo urinary elimination, the concentrations in the kidney cells being found 6 to 90 times higher than in the liver. This phenomenon explains in particular the nephrotoxicity reported within the framework of these intoxications, although dehydration due to gastrointestinal losses may also contribute to kidney injury [11,15]. »

-Recommendations regarding treatments, Table 1 and related text. Authors cannot recommend 1st, 2nd or whatever line treatments since RCT are absent. The % of deaths, taking into account the small number of each non-controlled studies, would not absolutely be statistically different even if compared in an overall Chi-squared test. 0% mortality in a clinical case series of ceftazidime + silybin looks must be cautiously interpreted, an additional patient who dies would give the expected mortality for the event.

Thank you for this relevant remark. We thus added a paragraph in the text to moderate our meanings, page 6 and 7 of the manuscript:

“Although no randomized controlled trials have been performed to date regarding the management of Amanita phalloides poisoning, the studies reported here provide survival data for patients associated with various drug classes after multidimensional statistical analysis. Thus, although no high level of evidence recommendation can be established yet, it is interesting to observe the survival trends of patients to propose an empirical therapeutic strategy.”

We have also added the group sizes in Table I.

-“To be excluded from care” (insulin, GH, glucagon, steroids); the evidence to deny them is the same as to propose the 1st and 2d line. Please comment that these treatments have been proposed in small series but these do not report better outcomes than other, which is far from stating “excluded”. There is no evidence to propose and no evidence to exclude, simply.

In addition to our previous comment, we have moderated our table by removing the section “to be excluded from care”. All treatments other than penicillins and silibinin are therefore grouped in the section “interest still to be demonstrated”.

-It is surprising that in the same way, authors state that antioxidants and aminoglycosides are promising with mortalities ranging 12-40%, truly??????? It looks like funny to test aminoglycosides and vancomycin among these patients that are at extremely high-risk of kidney injury... dialysis comes later.

This is correct. We completed the text page 7 of the manuscript:

“Apart from considerations of mechanism of action, it seems prudent to rule out the use of vancomycin or aminoglycosides in these patients at high risk of renal failure because of the nephrotoxicity of these molecules.”

-Treatments: besides the previous considerations they cannot be directly compared since, as occurring in uncontrolled studies, we do not know if patients were stratified by severity at presentation, time of ingestion, comorbidities, etc.

Data from Poucheret. et al. (2010) were analyzed using multidimensional statistic. We thus added this data page 6 and 7 of the manuscript:

“Although no randomized controlled trials have been performed to date regarding the management of Amanita phalloides poisoning, the studies reported here provide survival data for patients associated with various drug classes after multidimensional statistical analysis. Thus, although no high level of evidence recommendation can be established yet, it is interesting to observe the survival trends of patients to propose an empirical therapeutic strategy.”

-Summary regarding recommendation regarding treatments: penicilins or similars, along with silybin agglutinate the most positive reports. RCT are lacking, other therapies have not pointed at potential beneficial results in case series.

To clarify limitations of the reported data on amanitin antidotes, we included a summary regarding recommendations, page 6 of the manuscript:

“Overall, although randomized control study are lacking, penicillins or similar, along with silibinin agglutinate the most positive reports, and other therapies have not pointed at potential beneficial results in case series.”

-Acetylcisteyne is indicated with patients with fulminant liver failure (encephalopathy). For this reason it must be recommended.

We added it page 6 of the manuscript:

“N-acetyl-cysteine, which is indicated in patients with fulminant liver failure, must be recommended in this context.”

-Vitamin k and fresh frozen plasma (page 5, line 156) → they must not be routinely administered. When the hepatocyte is damaged vitamin k does not improve coagulation tests. Abnormal coagulation tests do not predict bleeding since liver failure both associates a decrease in pro and anticoagulant factors, so the final equilibrium is a “normal” coagulative state. Fresh frozen plasma has little impact on improving coagulative tests, which is not the same that coagulative status.

Thank you for these relevant comments. We thus moderated our sentence, page 6 of the manuscript:

“Also, inhibition of protein synthesis and hepatocyte damage may impair the production of coagulation factors, so intravenous administration of vitamin K and fresh frozen plasma may be necessary [35].”

-AKI in amanitin intoxication (page 3 line 81) → besides the potential nephrotoxic effects there’s a predominant component attributable to GI losses and pr-renal AKI.

Thank you for this comment. We thus added it page 5 of the manuscript:

“Acute kidney injuries frequently observed during these intoxications can be attributed to the nephrotoxic action of amanitins, to the gastrointestinal losses and to the previous state of kidneys.”

We would like to thank you for your helpful and constructive feedback concerning our manuscript and we hope that you will find it suitable for publication.

Reviewer 4 Report

General Comments

  1. Thank you for providing me with the opportunity to review this interesting article. This article was a narrative review of the toxic mechanisms, clinical manifestations, and supportive treatments of amanitin poisoning following the ingestion of amatoxic mushrooms, principally Amanita phalloides. An additional objective of the review included in its title was to present the latest advances in the study of amanitins that would permit "new effective therapeutics." Since no new and effective strategies were recommended for amanitin poisoning specifically, the title should be adjusted to clarify this and to inform the reader that the new therapeutics included the design of new amanitin-conjugated antibodies for cancer immunotherapy and not new therapeutics for amanitin poisoning. Consider: "Toxic effects of amanitins: repurposing toxicities toward new therapeutics."
  2. Although this was a narrative review and not a meta-analysis of case series of amanitin poisoning, the article still requires a methodology section which was not provided. How were the 56 references selected; by Internet search?  What were the search subject headings, the key words? What search engines were employed? What was the study period? What were the inclusion and exclusion criteria? Today, most reviews use the PRISMA recommendations to best describe their search and selection methodologies.
  3. Minor Comments. Page 4 of 9, line 142. Identify abbreviation ROS prior to its use. Page 5-6 of 9, Table 1. Provide reference numbers for the mortality rates for the other antidotes and therapies recommended beyond the first line recommendations. In the "Interest still to be determined" section of the Table, change à to a hyphen. 

Author Response

4th June 2021

Editorial

Toxins

Dear Editor,

Please find enclosed our revised manuscript entitled: “Toxic effects of amanitins: repurposing toxicities toward new therapeutics.” as a submission for publication in Toxins.

We would like to thank you for your helpful and constructive feedback concerning our manuscript. We discussed all the comments and - where appropriate - amended the manuscript accordingly as indicated. For better clarification, please find enclosed a reply letter that addresses all of the comments. In the manuscript, we highlighted all changes in red.

This work has not been nor will be submitted for publication in any other journal until you have taken your final decision.

We hope that you will find it suitable for publication.

Thank you for your cooperation.

Sincerely yours.

Reviewer #4:

  1. Thank you for providing me with the opportunity to review this interesting article. This article was a narrative review of the toxic mechanisms, clinical manifestations, and supportive treatments of amanitin poisoning following the ingestion of amatoxic mushrooms, principally Amanita phalloides. An additional objective of the review included in its title was to present the latest advances in the study of amanitins that would permit "new effective therapeutics." Since no new and effective strategies were recommended for amanitin poisoning specifically, the title should be adjusted to clarify this and to inform the reader that the new therapeutics included the design of new amanitin-conjugated antibodies for cancer immunotherapy and not new therapeutics for amanitin poisoning. Consider: "Toxic effects of amanitins: repurposing toxicities toward new therapeutics."

Thank you for this comment. We then modified the title page 1 of the manuscript:

“Toxic effects of amanitins: repurposing toxicities toward new therapeutics.”

  1. Although this was a narrative review and not a meta-analysis of case series of amanitin poisoning, the article still requires a methodology section which was not provided. How were the 56 references selected; by Internet search?  What were the search subject headings, the key words? What search engines were employed? What was the study period? What were the inclusion and exclusion criteria? Today, most reviews use the PRISMA recommendations to best describe their search and selection methodologies.

We thus included a methodology section page 1 of the manuscript:

“In this review, MEDLINE and PubMed databases were searched for relevant papers published in English between 1960 and 2021 using the following search terms : amanitin, amanita, antidotes, toxicity, toxicokinetics, liver, hepatocyte, hepatotoxicity, RNA polymerase, silibinin, penicillin. Studies providing information about mechanic explanation for amanitin toxicity, amanitin poisoning management and amanitin-conjugated antibodies were included for review. “

  1. Minor Comments. Page 4 of 9, line 142. Identify abbreviation ROS prior to its use. Page 5-6 of 9, Table 1. Provide reference numbers for the mortality rates for the other antidotes and therapies recommended beyond the first line recommendations. In the "Interest still to be determined" section of the Table, change à to a hyphen. 

These minor comments have been modified. References 38 and 40 are common to all lines, so we have not added an additional reference.

We would like to thank you for your helpful and constructive feedback concerning our manuscript and we hope that you will find it suitable for publication.

Round 2

Reviewer 1 Report

Thank-you for addressing the issues raised from the first review - most significant concerns were adequately addressed.  A few suggested edits and comments related to the newly added material.

Page 1, Line 43:  Studies providing....

Does "mechanic" mean "mechanistic"?  perhaps clarify

Page 6, lines 214-218: this sentence could benefit from a rewrite - generally unclear and "danger-associated molecular patterns" more specifically unclear as to meaning. 

Page 7, line 269 - "studies" not "study".

Pabe 7, line 276: "combination" not "combinaison".

Table 1: "vitamin E" not "vitamine E". 

Author Response

Thank-you for addressing the issues raised from the first review - most significant concerns were adequately addressed.  A few suggested edits and comments related to the newly added material.

Page 1, Line 43:  Studies providing....

Does "mechanic" mean "mechanistic"?  perhaps clarify

We apologize for this typo. We modified the text page 1 of the manuscript.

Page 6, lines 214-218: this sentence could benefit from a rewrite - generally unclear and "danger-associated molecular patterns" more specifically unclear as to meaning. 

We modified the sentence, page 5 of the manuscript:

“However in case of acute liver failure, treatment with high volume plasma exchange has been shown to increase liver transplant-free survival [36]. Beyond a potential detoxifying treatment, high volume plasma exchange would also allow the removal of danger-associated molecular patterns (molecules within cells that are a component of the innate immune response released from damaged or dying cells) involved in liver failure [36].”

Page 7, line 269 - "studies" not "study".

Pabe 7, line 276: "combination" not "combinaison".

Table 1: "vitamin E" not "vitamine E". 

Thank you for your comments. All the above comments have been taken into account and modified in the text.

We would like to thank you for your helpful and constructive feedback concerning our manuscript and hope that you will find it suitable for publication.

Reviewer 3 Report

I think that many references in Table 1 are lacking. Besides that, I have nothing else to say; authors have correctly adressed our questions. Congratulations for this nice review.

Author Response

I think that many references in Table 1 are lacking. Besides that, I have nothing else to say; authors have correctly adressed our questions. Congratulations for this nice review.

Thank you for this comment. No references are missing in Table I according to our methodological criteria cited on page 1 of the manuscript. Indeed, we only relied on the data of Enjalbert et al. (2012) and Poucheret et al. (2010) which are to date the largest studies in the literature comparing different treatments for Amanita phalloides intoxication.

Enjalbert, F.; Rapior, S.; Nouguier-Soulé, J.; Guillon, S.; Amouroux, N.; Cabot, C. Treatment of Amatoxin Poisoning: 20-Year Retrospective Analysis. J. Toxicol. Clin. Toxicol. 2002, 40, 715–757, doi:10.1081/CLT-120014646.

Poucheret, P.; Fons, F.; Doré, J.C.; Michelot, D.; Rapior, S. Amatoxin Poisoning Treatment Decision-Making: Pharmaco-Therapeutic Clinical Strategy Assessment Using Multidimensional Multivariate Statistic Analysis. Toxicon 2010, 55, 1338–1345, doi:10.1016/j.toxicon.2010.02.005.

We would like to thank you for your helpful and constructive feedback concerning our manuscript and hope that you will find it suitable for publication.
